# IoT System for Gluten Prediction in Flour Samples Using NIRS Technology, Deep and Machine Learning Techniques

Oscar Jossa-Bastidas [1,*], Ainhoa Osa Sanchez [1], Leire Bravo-Lamas [2] and Begonya Garcia-Zapirain [1]

[1] eVIDA Research Group, University of Deusto, 48007 Bilbao, Spain; ainhoa.osa.sanchez@deusto.es (A.O.S.)
[2] Food Technology Department, Leartiker S. Coop., Xemein Etorbidea 12, 48270 Markina-Xemein, Spain; lbravo@leartiker.com
[*] Correspondence: oscar.jossa@deusto.es

**Abstract:** Gluten is a natural complex protein present in a variety of cereal grains, including species of wheat, barley, rye, triticale, and oat cultivars. When someone suffering from celiac disease ingests it, the immune system starts attacking its own tissues. Prevalence studies suggest that approximately 1% of the population may have gluten-related disorders during their lifetime, thus, the scientific community has tried to study different methods to detect this protein. There are multiple commercial quantitative methods for gluten detection, such as enzyme-linked immunosorbent assays (ELISAs), polymerase chain reactions, and advanced proteomic methods. ELISA-based methods are the most widely used; but despite being reliable, they also have certain constraints, such as the long periods they take to detect the protein. This study focuses on developing a novel, rapid, and budget-friendly IoT system using Near-infrared spectroscopy technology, Deep and Machine Learning algorithms to predict the presence or absence of gluten in flour samples. 12,053 samples were collected from 3 different types of flour (rye, corn, and oats) using an IoT prototype portable solution composed of a Raspberry Pi 4 and the DLPNIRNANOEVM infrared sensor. The proposed solution can collect, store, and predict new samples and is connected by using a real-time serverless architecture designed in the Amazon Web services. The results showed that the XGBoost classifier reached an Accuracy of 94.52% and an F2-score of 92.87%, whereas the Deep Neural network had an Accuracy of 91.77% and an F2-score of 96.06%. The findings also showed that it is possible to achieve high-performance results by only using the 1452–1583 nm wavelength range. The IoT prototype portable solution presented in this study not only provides a valuable contribution to the state of the art in the use of the NIRS + Artificial Intelligence in the food industry, but it also represents a first step towards the development of technologies that can improve the quality of life of people with food intolerances.

**Keywords:** IoT; deep learning; gluten; near-infrared spectroscopy; machine learning; feature selection; flour samples

## 1. Introduction

Gluten is a protein formed by prolamin and glutelin that is found at different percentages in wheat, barley, rye, and possibly in some cultivars of oats due to cross-contamination [1]. Gluten gives elasticity to dough and glue-like consistency and spongy properties to pasta, bread, and cakes because of the interaction between prolamin and glutelin in presence of water and energy. However, prolamin is highly composed of glutamine and proline amino acids, which are known for being indigestible. They also prevent a complete enzymatic breakdown of gluten in consumers' intestines, increasing the concentration of toxic oligopeptides [2]. Celiac disease (CD), one of the five major illnesses associated with gluten [3], is a multifactorial autoimmune disease triggered in individuals genetically predisposed to gluten intake, which has been on the rise in the past 15–20 years [4,5]. The resulting small intestinal inflammatory process is accompanied by the production of specific antibodies against gliadin (prolamin of wheat) and tissue transglutaminase (TG),

leading to a variety of gastrointestinal and extraintestinal manifestations of a wide range of severity [4,6]. Despite the prevalence of CD rising to 1% worldwide; with some studies reporting a fourfold increase in the past 50 years, CD remains an often underdiagnosed condition [7–9].

With the number of people suffering from disorders caused by gluten increasing, Health Authorities have approved label regulations to protect consumers. Europe Commission Implementing Regulation (EU) No 828/2014 [10] indicates that gluten-free products should contain <20 mg/kg of gluten and very low gluten products <100 mg/kg. While these levels of detection can be quantified using highly sensitive methods, such as instrumental analytical techniques (capillary electrophoresis, PCR, QC-PCR, RP-HPLC, LC-MS, and MALDI-TOF-MS), they also imply elevated costs and specialized training [11–13]. The latest methodologies for quantifying gluten are based on DNA instead of protein extraction. Although they allow gluten quantification, they are expensive and have limitations with highly processed foods [12].

With gluten concentration control being essential for the food industry, the need for reliable, simple to use, and affordable analytical methods has led to the majority of commercialized gluten detection kits being based on immunochemical methodologies (ELISA). In fact, AOAC Official Method has validated one of these kits to measure gluten in various types and forms of food [14]. Although some authors have detected limitations and variations of 20 ppm among results using ELISA commercial kits for gluten quantification on heated and processed food [12,13,15], others have noticed some improvements when applying the methods to dairy products [16] and plant seeds [17], with results that allow quantification until 5.5 ppm. The ELISA method is quicker and easier to perform than instrumental techniques, however, it takes time (from 30 min to 2.5 h [13]) and laboratory equipment is necessary. All aforementioned analytical methods work having the analyte (protein) as the target, however, in the last few years, indirect analysis based on AI methods has been proposed more frequently [18]. They work using large databases, offering quick and cheap solutions to the food industry [19].

Near-infrared spectroscopy (NIRS) sensors have been proposed in some studies to analyze different proteins, including gluten. In a recent study, a Fourier transform infrared spectrometer technique was used to determine the total wheat protein and the wet gluten present in wheat flour. The authors obtained an external validation of 82% for $R^2$, and concluded that it is possible to predict the gluten protein content of wheat flour using NIRS [20]. Additionally, in a recent systematic review, it was concluded that the NIRS is an excellent technique for analyzing the quality and safety of flour and it allows non-destructive analysis [21]. Similarly, a non-invasive and quick method was developed to determine the authenticity of vegetable protein powders and classify possible adulterations using NIRS and chemometric tools [22]. In this study were investigated three potential powder adulterants: whey, wheat, and soy protein. The authors used the one-class partial discriminant analysis for the authentication and the partial least squares discriminant analysis (PLS-DA) for the classification of the adulterants. In total 144 adulterated samples and 14 pure plant-based protein powder samples were analyzed. They achieved 100% sensibility and specificity in the prediction set in the proposed PLS-DA model to authenticate pure plant-based protein powder samples and classify adulterants [22]. In another study, the NIRS was used in the 900–2250 nm wavelength range to assess the protein content in potato flour noodles in a non-destructive manner. The performance of the model was evaluated through partial least squares regression (PLSR), and the results indicated a high degree of accuracy with an $R^2$ value of 0.8925 and RMSE value of 0.1385% for the prediction set [23]. In another investigation, the NIRS technology was used to predict the purity of the flour PLSR using six samples of authentic almond flour. The study utilized three different NIRS devices: a MicroNIR$^{TM}$ working in the 950–1650 nm wavelength range; a DLPR NIRscanTM Nano working in the 900–1700 nm wavelength range; and a NeoSpectra FT-NIR operating in the 1350–2500 nm wavelength range. The classification



results achieved were 100% sensitivity and more than 95% specificity, and with $R^2$ value of 0.90 [24].

Machine Learning (ML) and Deep Learning (DL) methods have been used in many application fields, including image recognition, audio classification, and lately natural language processing [25–27]. These Artificial Intelligence (AI) algorithms have also been combined with the potential of NIRS technology to solve problems in the food industry, in most cases related to the evaluation of food samples. In a recent study NIRS technology and ML models were used to evaluate the quality traits of sourdough bread flour made from six different flour sources [28]. The results showed that the ML models were able to classify the type of wheat used for the flours with an Accuracy and precision of 96.3% and 99.4% respectively by using the NIRS and a low-cost electronic nose. In another study, a rapid and non-destructive classification of six different Amaranthus species was conducted by using the Visible and Near-infrared (Vis-NIR) spectra in the ranges between 400–1075 nm and chemometric approaches [29]. The authors evaluated four different preprocessing methods to detect the optimal preprocessing technique with the highest classification Accuracy. The different preprocessing and modeling combinations showed classification accuracies from 71% to 99.7% after cross-validation. The combination of Savitzky-Golay preprocessing and Support Vector Machine (SVM) showed a maximum mean classification Accuracy of 99.7% for the discrimination of Amaranthus species. The authors concluded that Vis-NIR spectroscopy, in combination with appropriate preprocessing and AI methods, can be used to effectively classify Amaranthus species. Similarly, a portable low-cost spectrophotometric device was designed to classify 9 different food types of powder or flake structures. The device worked in the Vis-NIR region, and it employed ML algorithms for the data classification [30]. 18 features belonging to each sample were collected in the optical spectral region in the range between 410–940 nm. The SVM and the Convolutional Neural Network (CNN) achieved an Accuracy of 97% and 95%, respectively. A recent investigation used feature selection techniques to determine the most important wavelength ranges for gluten detection in flour samples. The highest Accuracy obtained was 84.42% by selecting the 1089–1325 nm wavelength range and using the Random Forest classifier [31].

The main objective of this study and the contribution to the state of the art was to develop an innovative, rapid, and budget-friendly IoT system using NIRS technology, ML, and DL algorithms for the detection of the presence or absence of gluten in three types of flour samples.

The manuscript is organized as follows. Section 2 explains the materials and methods employed in the study, including the preparation and collection of the data, the implementation of the hardware and software, and the theory of the methods used. In Section 3, the results are presented and analyzed in three subsections that include the hyperparameter tuning methodology used for the ML and DL methods, and their classification results. Finally, Section 4 presents the discussion and conclusions, highlighting the major findings of the study.

## 2. Materials and Methods

### 2.1. Data Preparation

Among the food that could respond adequately to NIRS technology, we selected commercial flours. Flour presents a homogenous color and is easy to work with.

We acquired commercial flours of rye, corn, and oat in February 2022 from two different Spanish brands, as seen in Table 1. They were analyzed in the Food Technology Department of Leartiker S. Coop twice using the sandwich ELISA commercial kit (Gluten (gliadin), Biosystem, Spain). High concentrations of gluten were quantified in rye flour (average of 22.4 g/kg), while very low concentrations were found in the oat flour (13.4 ppm) and corn samples (3.0 ppm) from El Granero, as seen in Table 1. However, the oat flour samples from El Alcavaran were quantified with 113 ppm more gluten compared with El Granero. Thus, the corn flour samples from two brands and the oat flour samples from El Granero can be labeled as gluten-free products (<20 ppm [10]).

**Table 1.** The specifications and natural gluten content of the commercial flours used for the preparation of the samples.

| Type of Flour | Brand | Specifications | Natural Gluten Content (ppm) |
|---|---|---|---|
| Rye | El Alcavaran | Whole. Set no. 30620184/189/21 | 33,650 ± 4413.0 |
| Rye | El Granero | Organic production. Whole. Set no. HC091231 | 11,108 ± 409.41 |
| Corn | APASA | Set no. 024228 | <LOD |
| Corn | El Granero | Organic production. Whole. Set no. HM281031 | 3.0210 * ± 0.1909 |
| Oat | El Alcavaran | Organic production. Whole. Set no. A-40320172-040/21 | 126.02 ± 89.095 |
| Oat | El Granero | Organic production. Whole. Set no. HAI161231 | 13.402 ± 0.4755 |

LOD, limit of detection. *, < LQD (limit of quantification, 4 ppm).

We then adulterated each flour with commercial gluten (El Granero, Spain; set no. GT301131) for the training of the NIRS sensors. Per each kg of flour, 100 g of gluten was added and mixed (Thermomix$^R$ TM6, Vorwerk, Wuppertal, Germany) for 15 min. The mixing procedure was validated in a previous step, where corn flour from the APASA brand was adulterated to 20 ppm of gluten. Afterward, we took 4 different samples from the cup (upper, bottom, two intermediate, and lateral zones) and analyzed them twice using the ELISA commercial kit (Gluten (gliadin), BioSystem, Barcelona, Spain). Finally, we divided each flour's adulterated samples with concentrations of 0 (control), and 100 g of gluten per kg into subsamples and stored them in identified and airless zip bags that were sent to the facilities of the University of Deusto.

### 2.2. Implementation

Once the samples were prepared, we proceeded with the data collection, data analysis, and creation of the ML and DL prediction models. Figure 1 shows the hardware and software architecture used to collect the data and the communication among them.

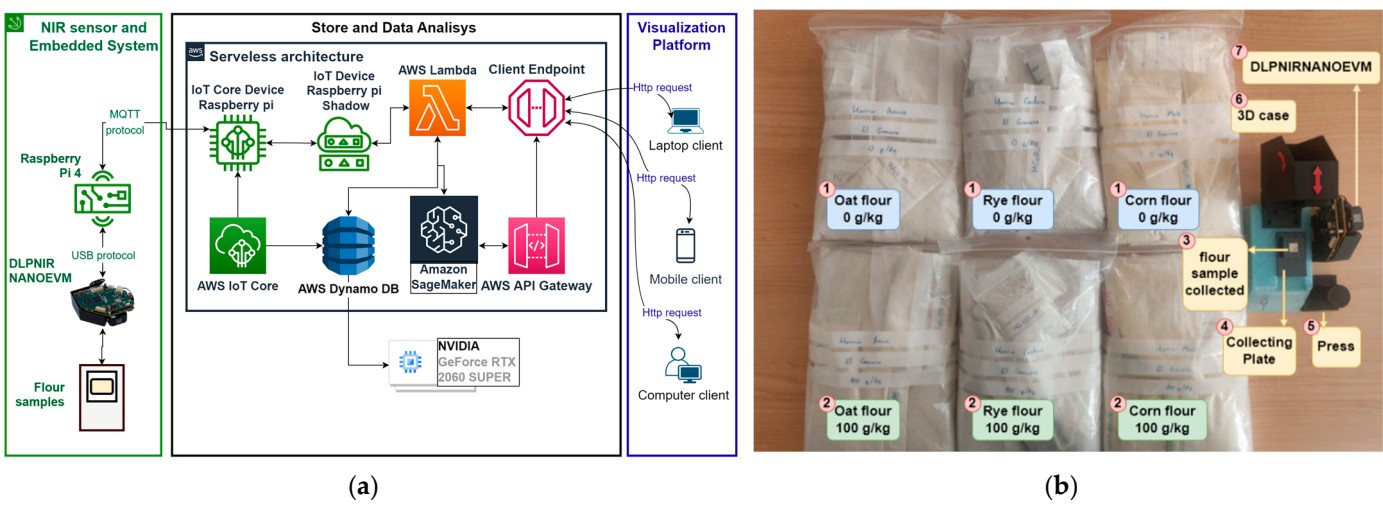

(**a**)  (**b**)

**Figure 1.** Software architecture, hardware, flour samples, and 3D mechanical system used. (**a**) Serverless architecture. (**b**) Flour samples, NIRS sensor, and IoT prototype portable solution.

Hardware and Software Description

Below is the explanation for each stage of the Hardware and software architecture shown in Figure 1a.

- NIRS sensor and embedded system: as seen in Figure 1a, the IoT prototype portable solution is composed of a Raspberry Pi 4 (Raspberry Pi is a small and low-cost computer, which uses a screen a keyboard, and a mouse and can be used by people of all ages to learn how to program in programming languages as Scratch and Python [32]) and the DLPNIRNANOEVM a compacted evaluation module used for NIRS [33]. The DLPNIRNANOEVM sensor works in the 900–1700 nm wavelength range with a resolution of 4 nm, so one measure provides a total of 228 variables. The output of the sensor was the intensity. The DLPNIRNANOEVM sensor was connected to the Raspberry Pi 4 using the USB protocol and Python scripts were designed for collecting data from the flour samples. To activate the sensor, we created an HTTP endpoint for receiving different requests using JSON files. This endpoint receives different information, including the sensor's name and id, as well as the action to execute, with different parameters such as the sensor's state and the duration of data collection. The endpoint then enables an AWS lambda function that changes the status of the shadow in the AWS IoT microservice, which initiates data collection using the Message Queuing Telemetry Transport (MQTT) protocol (MQTT is a standard messaging protocol based on publish/subscribe messaging communication which is ideal for connecting remote devices [34]).
- Data storing and data analysis: the data was received by AWS IoT and forwarded to AWS Lambda, which had several functionalities: managing the logic requests to the database, making the ML and DL predictions, and exposing the endpoints to the final user (client). The data was stored using AWS DynamoDB. The full explanation of this architecture is given in [35].
- For the data analysis, we employed ML and DL algorithms programmed using Python programming language, version 3.9. On the one hand, the ML techniques were trained using the ml.m4.xlarge instance available in AWS sagemaker [36] equipped with 4vCPU and 16 GiB. On the other hand, the DL models were trained using a local machine equipped with an NVIDIA GeForce RTX 2060 SUPER graphic card with 8 GB of VRAM memory and 16 GB of shared GPU memory. We trained the DL algorithms in an eVida local machine to take advantage of the power of the graphic card but also because this made it easier to manipulate the system files, something useful for a custom tuning methodology later explained.
- Visualization platform: it was designed using the Django framework and is communicated with the AWS platform using HTTP requests. The functionalities of this platform are the visualization of gluten measures, the collection of new samples, and the visualization of new predictions. It is worth mentioning that the explanation of the visualization platform is out of the scope of this study, therefore, we do not go into details.

To make a new prediction the IoT prototype portable solution takes between 30–60 s approximately, starting from the flour sample collection until the data is predicted with the presence or absence of gluten and visualized in the platform.

### 2.3. Data Collection Procedure

In Figure 1b, the blue and green boxes show the different types of flour with 0 g/kg and 100 g/kg of gluten concentration respectively. The data collection was carried out over approximately 2 months by 4 researchers of the eVida research team. We followed an internal protocol to ensure a correct data collection procedure, which is explained below:

1. For each new measurement, the operator had to change the collecting plate (grams capacity $\approx$ 400 mg) (4 in Figure 1b) and then take the flour samples from different locations of the bag (1 or 2 in the same figure). Hence, he/she scooped flour from the top, bottom, center, and lateral sides of the bag to randomize the data collection process as much as possible. Finally, the operator put the samples in the collecting plate (4 in Figure 1b).

2.    Once the sample was on the collecting plate, the operator smashed the flour trying to keep it on a smooth surface. This was because, after some experiments, we realized that when the surface was not smooth, the data collection was not consistent.
3.    To avoid cross-contamination of the samples, the operator had to wear different gloves when collecting the data from different flour types. Furthermore, it was necessary to use different spoons and collecting plates for each type of flour. The gloves were thrown away at the end of the day.
4.    The time collection per sample was approximately 30 s, during this time window, the DLPNIRNANOEVM sensor measured the exposed sample and forwarded the data to the AWS platform.
5.    All the samples were collected with the same sensor and embedded system. Therefore, to measure the data it was necessary to design and print a 3D mechanical system. On the right in Figure 1b the 3D mechanical system is shown, it is composed of a 3d black case at the top (it contains the DLPNIRNANOEVM inside) and the blue box at the bottom (it contains the Raspberry Pi 4). It was used to keep the sensor rigid during the measuring process, but also to collect the data in a dark environment.

### 2.4. Classification Framework

Figure 2 shows the pipeline we followed for the creation of a model able to predict the presence of gluten in the flour samples. Please note that when we refer to the "absence of gluten", we are specifically alluding to flour samples that have a doping level of 0 g/kg of gluten.

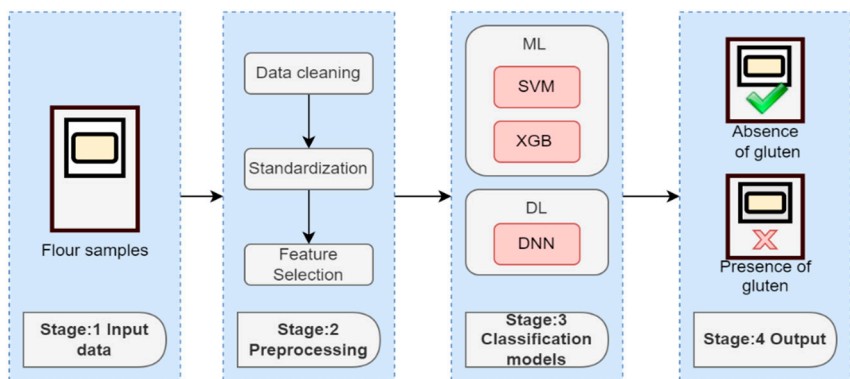

**Figure 2.** Hardware and software architecture.

### 2.4.1. Input Data

In the present study, we focused our efforts on a dichotomic case. Thus, we collected samples from 3 different types of flour (rye, corn, and oats) and doped them with 0 g/kg and 100 g/kg of gluten concentration as explained before.

The sensor provided 228 variables (intensity) for each wavelength from 900 nm to 1700 nm. We collected a total of 12,053 observations and split the dataset as follows: 70% (8438 observations) for the training set, 15% (1807 observations) for the validation set, and 15% (1808 observations) for the testing set. It is important to note that the testing set was kept apart from the other ones from the very beginning of the collection process by using different bags. Thus, it was totally unknown for the ML and DL algorithms.

The three different types of flour samples rye, corn, and oat from the training dataset are represented by the mean of observations in Figure 3. The mean of the flour samples doped with 0 g/kg is, on average, higher than the mean of the samples doped with 100 g/kg. However, for low (near 900 nm) and high (near 1700 nm) wavelengths the means overlap.

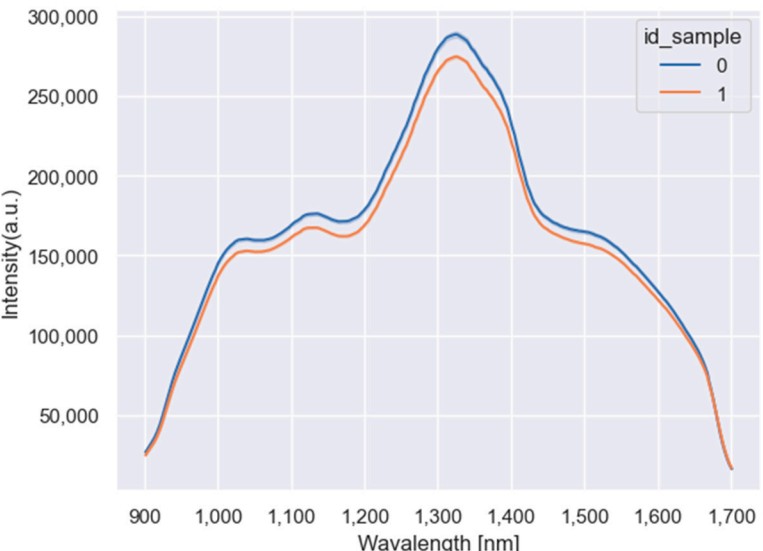

**Figure 3.** Mean of the flour samples doped with 0 g/kg (blue line) and 100 g/kg of gluten (orange line).

2.4.2. Preprocessing

The preprocessing stage was divided into three phases: the cleaning of the data, the standardization, and the feature selection procedure.

- Data cleaning: Despite the good quality of the data provided by the DLPNIRNA-NOEVM sensor, we checked that it met the following requirements. First, we checked the valid data, looking at whether the variable names and values met the required formats. Second, we checked the complete data, looking for NaN values and replacing them with valid values. Third, we checked the consistent data, looking at the outlier values in the dataset. Finally, we checked the unique data deleting the duplicate values.
- Standardization: once we cleaned the data, the next step was applying the standardization of the data. We applied row standardization, given the fact that all the columns had the same unit. During this process, the variables were standardized by removing the mean and scaling to unit variance [37]. The standard score is given by (1).

$$z = \frac{x - u}{\sigma} \tag{1}$$

where $x$ is the wavelength value, $u$ is the mean of all the wavelength variables and $\sigma$ is the standard deviation.

- Feature selection: We selected the 3 wavelength ranges to train the ML and DL algorithms: 1089–1325 nm; 1239–1353 nm and 1422–1583 nm; and the whole spectrum 900–1700 nm, based on our previous study [31]. The purpose was to corroborate the possibility of predicting the presence or absence of gluten in the flour by only selecting some of the wavelength variables.

2.4.3. Classification Models

2.4.3.1. Support Vector Machine

SVM is a supervised learning method used for regression, classification, outlier detection, and feature selection problems [38]. The main objective of an SVM algorithm is to create an optimal hyperplane that separates the classes as much as possible. In the case of a bi-dimensional space, the hyperplane is a line; in tridimensional space, the hyperplane is 2-dimensional planes; and so on the SVM creates n-dimensional $R^{n-1}$ planes, where n is the dimension or number of features [39]. A kernel function is needed to map the data.

Given a set of $n$ observations with $x$ representing the training data and $y$ the class of the label, as seen in (2).

$$S = \{(x_1, y_1), \ldots, (x_i, y_i)\} \tag{2}$$

The decision function for nonlinear data of the algorithm is given by (3) [39]. Where $m$ is the bias parameter and $\alpha$ determines the maximal margin classifier, a parameter related to the input vector.

$$f(x) = sgn(\sum_{i=1}^{N} \alpha_i y K_i(x_i.x) + m) \tag{3}$$

where $K$ is the kernel function. In this study, we used the Polynomial, Radial basis function, and Sigmoid kernels.

### 2.4.3.2. Extreme Gradient Boosting (XGBoost)

XGBoost algorithm is an implementation of the gradient-boosted decision trees designed for speed and higher performance. The term "Gradient Boosting" was used by Friedman in 2001 and is applied to structured and tabular data [40]. There are C++, Python, R, Java, and more implementation libraries of this algorithm proposed by Tianqi Chen [41].

The XGBoost takes advantage of the second-order Taylor expansion of the loss function and adds a regularization term to balance the complexity of the model and the decline of the loss function. Equation (4) is used to calculate a prediction having a dataset with $n$ examples and $m$ features.

$$\hat{y}_i = \phi(x_i) = \sum_{k=1}^{K} f_k(x_i), \ f_k \ \epsilon \ F \tag{4}$$

where $F = \{f(x) = w_{q(x)}\}(q : R^m \rightarrow T, w \in R^T)$ is the spacing of the trees, $q$ is the structure of each tree and $T$ is the number of leaves of the tree. Therefore, each $f$ is an independent tree structure. The regularized objective can be minimized as follows in (5).

$$L(\phi) = \sum_i l(\hat{y}_i, y_i) + \sum_k \Omega(f_k) \tag{5}$$

where $\Omega(f) = YT + \frac{1}{2}\lambda||w||^2$.

In this case, $l$ is the convex of the loss function and $\Omega$ is the penalization of the complexity of the model. With the aim of improving the objective $i$ instance and $t$ iteration are added and using a second-order approximation obtaining (6) [41].

$$L^{(t)} \simeq \sum_{i=1}^{n} \left[l\left(y_i, \hat{y}_i^{(t-1)} + g_i \ f_t(x_i) + \frac{1}{2}h_i f_t^2(x_i)\right)\right] + \Omega(f_t) \tag{6}$$

where $g_i = \partial_{\hat{y}} l\left(y_i, \hat{y}_i^{(t-1)}\right)$ and $h_i = \partial^2_{\hat{y}^{(t-1)}} l\left(y_i, \hat{y}_i^{(t-1)}\right)$ are first and second order gradient statistics on the loss function. Removing the constant terms and defining I_j = { i | q(x_i)= j} as the instance of $j$ [41]. Also, defining $w_j^* = -\frac{\Sigma_{i\in Ij} \ gi}{\Sigma_{i\in Ij} \ h_i+\lambda}$ finally (7) is obtained. This can be used as a scoring function to measure the quality of a tree.

$$L^{(t)}(q) = -\frac{1}{2}\sum_{j=1}^{T} \frac{\left(\Sigma_{i\in Ij} \ gi\right)^2}{\Sigma_{i\in Ij} \ h_i + \lambda} + \gamma T \tag{7}$$

### 2.4.3.3. Deep Neural Network

The functionality of DNN is mainly described in two processes: the forwarding and the backpropagation process. The Deep feedforward networks, also known as multilayer perceptron (MLP), are the quintessential DNN models and have the objective of approximating a function $f^*$ [42]. These models are called feedforward because the computation

starts from the input $x$, going through the intermediate computations until obtaining the result of the output $y$. The feedforward networks are represented by a composition of multivariate functions.

$$f = g \circ fk \circ \ldots \circ f2 \circ f1 \tag{8}$$

where $k$ is the last hidden layer and $g$ is the output function. The function maps its inputs into the outputs $f: Rm \rightarrow Rs$; where $m$ is the dimension of the $x$ inputs and $s$ is the dimension of the output. Each hidden layer is composed of multivariate functions.

$$f_i(x) = a(w_i x + b_i) \tag{9}$$

Replacing (9) in (8).

$$f(x) = g(a(\ldots a(w_2 a(w_1 x + b_1) + b_2) \ldots + b_K)) \tag{10}$$

where $f_i$ are the hidden layer functions, $w$ is the weight, $b$ is the bias and $a$ is the output activation. Each linear combination plus the bias produces the output of the node.

As we said before, the forward process takes the input $x$ and propagates up to the hidden layer until obtaining the $\hat{y}$. During this process is produced a scalar cost $J(\theta)$. The backpropagation propagates then the information backward through the network, with the purpose of calculating the gradient [42]. Suppose that $x \in R_m$, $y \in R_s$, $f$ and $g$ both map from real number to real number $R_m \rightarrow R_s$, and $f$ maps $R_n$ to $R$. If $y = g(x)$ and $z = f(y)$; $z = f(g(x))$, then using the chain rule.

$$\frac{\partial z}{\partial x_i} = \sum_j \frac{\partial z}{\partial y_i} \frac{\partial y_i}{\partial x_i} \tag{11}$$

$$\nabla_x z = \left( \frac{\partial y}{\partial x} \right)^\top \nabla_y z \tag{12}$$

In (12), (11) is rewritten in vector notation, where $\frac{\partial y}{\partial x}$ is the $n \times m$ Jacobian matrix of $g$. It is possible to obtain the gradient of a variable $x$ by multiplying the Jacobian matrix by a gradient $\nabla_y z$. Thus, the backpropagation is the result of multiplying the Jacobian gradient for each operation in the graph [42].

2.4.3.4. Hyperparameter Tuning Methodology for DNN

For the DNN hyperparameter tuning we proposed a methodology based on the operation principles of the Hill climbing and the Grid Search methods. The idea of the proposed methodology is to tune each hyperparameter in the desired order and to tune each hyperparameter individually looking to obtain the higher metrics for each iteration. We decided to use this methodology because the literature lacks a method interested in the order in which the hyperparameters should be tuned. It is worth mentioning that we are not suggesting a general specific order of the hyperparameters in this study, however, the methodology is designed to investigate the hyperparameter order for future investigations. Likewise, we still do not refer to the methodology as a new framework, because it lacks an optimal implementation that cares about time, and, at this point, in time it is only implemented using sequential logic.

Table 2 shows the pseudocode for the methodology proposed. The following abbreviations were used to simplify the process.

**Table 2.** Pseudocode of the proposed tuning methodology.

1. Input: $HM, f_{hm}, m$
2. Initialize $MV = (r)$
3. for $hm_i$ in $HM$
    3.1. for $mv_k$ in $MV$
        3.1.1. for $hmij$ in $hm_i$
            3.1.1.1. train_DNN $(mv_{kj}, hmij)$
            3.1.1.2. $m\_val$ <- calculate $(m)$
            3.1.1.3. $s[s_{kj}]$<- set_score($m\_val$)
        3.1.2. $(mv_{11}, mv_{12}, \dots, mv_{f_{hm_i}j})$ <- $select\_max_{f_{hm_i}}(s_{11}, s_{12}, \dots s_{1j})$
    3.2. $MV$ <- $(mv_{11}, mv_{12}, \dots, mv_{kj})$); $k = 0$.
4. Output: $fmv$ <- $MV$

- Hyperparameters Matrix ($HM = (hm_{ij})$; $1 \leq i \leq l$, $1 \leq j \leq n$): it is a matrix of hyperparameters and its values. The hyperparameters will be tuned in the row ($i$) order indicated.
- Filter ($f_{hm} = [f_{hm_i}]$): it is a vector (for each hyperparameter) of thresholds that limit the number of models that pass to the next $hm_i$ iteration (See Table 2, step 3.1.2).
- Metrics ($m$): it is a vector of the metrics to evaluate the performance of the models. The metrics will determine the score for each model.
- Model values ($MV = (mv_{kj})$; $1 \leq k \leq f_{hm_i}$, $1 \leq j \leq n$): it is a matrix that contains the best hyperparameter values for each model selected during each $hm_i$ iteration. (See Table 2, step 3.2).
- Final model values ($fmv$): it is a vector of the hyperparameter values for the best model selected at the end of the process.
- Score($s$): it is a vector of the scores for each $ij$ iteration. The vector is rewritten for each $j$ iteration.
- Random vector ($r$): it is a random vector.

where $i$ is the hyperparameter iterated, $j$ is the respective values, $l$ is the number of hyperparameters, and $n$ is the number of hyperparameter values. Furthermore, we followed the matrix notation: the matrixes are in capital letters and bolded, the vectors are in lowercase and bolded, and the scalars are in lowercase.

The algorithm takes as inputs the $HM$ matrix and the $f_{hm_i}$, $m$ vectors. The values of the $HM$ matrix are randomly initiated. The algorithm then is going to look for the bests $MV$ model values for each hyperparameter $hm_i$; for this, the algorithm is going to train the DNN algorithm, calculate the metrics and set a score for each model. The set_score method used in this study consists of giving the models a rating number for the metrics $m$ selected, however, it could be changed for any other method to score the models. Once all the values have been tested, the algorithm sorts the $s$ vector and selects the number of $f_{hm_i}$ models. The $mv$ vector that contains the values is assigned to the $MV$ matrix and the $k$ value is restarted to zero (i.e., the algorithm only keeps the $mv$ values of the current iteration and removes the values of past $hm_i$ iterations). For the second iteration, the procedures from 3.1.1 to 3.1.2 will be executed again, but this time considering the best $mv$ values from the previous $hm_i$ iteration. Finally, the vector of values at the end of the $hm_i$ iterations will be assigned to the $fmv$ vector and those are the values tuned at the end of the algorithm. It is worth noting that the last values $f_{hm_i} = f_{hm_l} = 1$ because only one model is selected.

For a better understanding, we also graph the tuning process as seen in Figure 4. The red color represents the inputs while the blue one represents variables calculated during the process. In Figure 4 each hyperparameter $hm_1$, $hm_2$, $\dots$, $hm_l$, of the matrix $HM$ is represented as a green box. During each $hm_i$ iteration the $MV$ matrix is updated depending on the score calculated, the filters $f_{hm_1}$, $f_{hm_2}$, $\dots, f_{hm_l}$ and the metrics $m$. In Figure 4, can be noted that the order of the hyperparameters matters. Thus, during the first iteration the $hm_1$ hyperparameter is tuned, subsequently, in the second iteration, the $hm_2$ hyperparameter is tuned, using the hyperparameter values found in the previous iteration

$hm_1$. Therefore, the algorithm will tune the hyperparameters during each iteration until the end of the process. In Figure 4, the Experiment Numbers ($EN_i$) correspond to the number of executed experiments for each hyperparameter. $EN_i$ was multiplied by three, because we repeated each training three times, and for each result we computed the mean of the training. This was done to reduce the randomness in the results.

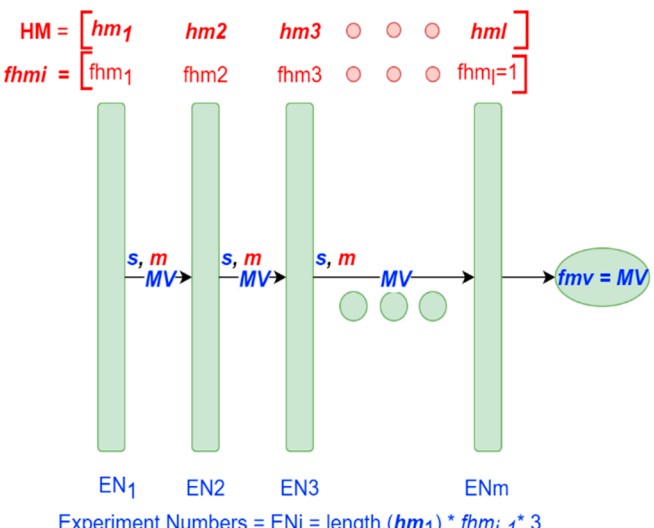

**Figure 4.** Graph procedure of the proposed tuning methodology.

2.4.4. Output

The output ŷ for the ML and DL models is the label indicating if the flour sample contains gluten or not. In this study "1" represents the presence of gluten and "0" represents its absence. To measure the performance of the classifiers, we calculated the metrics Accuracy and F2-score. We are particularly interested in the F2-score since it gives more weight to recall than the precision, which is particularly important considering overlooked False Negatives ($FN$) are more detrimental than False Positives ($FP$) in the prediction of the presence of gluten. For instance, wrongly predicting the absence of gluten in the flour will cause the user to consume gluten and suffer from potential adverse health effects. The equations for the Accuracy and F2-score are given by (13) and (14) respectively.

$$\text{ACC} = \frac{TP + TN}{TP + FP + TN + FN} \tag{13}$$

$$F2 = \frac{TP}{TP + 0.2FP + 0.8FN} \tag{14}$$

The true positives ($TP$) represent the flour samples that were correctly predicted with the presence of gluten, while the true negatives ($TN$) represent the samples that were correctly predicted with the absence of gluten. The $FP$ represents the samples that were predicted with the presence of gluten but did not have it, and finally, the $FN$ represents the samples that were predicted with the absence of gluten but actually had it.

**3. Results**

This section provides the results obtained for the ML and DL algorithms. We focused our effort on trying to get the best models for each wavelength range proposed for both ML and DL methods. The ML results for each wavelength range model were compared because they were trained under similar conditions (same tuning method, same machine, and the same amount of training). Likewise, the DL methods were compared with the results for each wavelength range, and we show the results using the proposed methodology for hyperparameter tuning. However, despite employing a similar amount of training for the

ML and DL models, it is not possible to make a fair comparison between them, because they were trained using different machines. It is important to mention that the sections about hyperparameter tuning 3.1 and 3.2 only present the results obtained for the validation set.

### 3.1. Machine Learning Hyperparameter Tuning

The AWS Sagemaker hyperparameter tuning tool was used to adjust the hyperparameters of the ML algorithms. The hyperparameters tuning experiments for SVM and XGBoost were executed 10 times, with each execution having 100 iterations. The selected tuning method was the Bayesian Optimization process available in AWS Sagemaker [43].

The Bayesian optimization algorithm differs from grid search or random search since it considers all the historical evaluations [44]. It can be solved mathematically as follows by Equation (15) [45,46]. The objective function is defined in the domain of $X$; $f : X \rightarrow R$.

$$x^* \in arg \, max_{x \in X} \, f(x); \tag{15}$$

#### 3.1.1. SVM

We considered the following hyperparameters for this study: C is the regularization parameter; gamma defines the reach of the influence of a single training, with low values and high values meaning far and close, respectively; and the kernel function that transforms the input data in a high dimensional space. The ranges of the parameters used for the tuning are shown in Table 3.

**Table 3.** Search space of the SVM method.

| Hyperparameters | Lower Limit | Upper Limit | Kernel Types |
| --- | --- | --- | --- |
| C | 0.000001 | 1000,000 | - |
| kernel | - | - | poly, rbf, sigmoid |
| gamma | - | - | scale, auto |

Below are the results of the hyperparameter tuning of the ML methods. The results revealed the impact of applying hyperparameter optimization in the different wavelength ranges. After the optimization procedure, the maximum Accuracy and F2-score were approximately 95% for the first two wavelength ranges and 80% for the third one. Furthermore, the same figure suggests that the models for the 900–1700 nm wavelength range were overfitting.

Additionally, Figure 5 shows the bivariate distribution for C and training time. Therefore, a darker red color indicates more concentration or values repeated while the light red color represents the opposite case. From this figure, it can be appreciated that the Bayesian optimization considers the historical results during the training as we expected. This means that, after the first execution iteration where the hyperparameter combinations were random, the Bayesian optimization algorithm found regions where it executed experiments more frequently. This behavior was repeated for the four wavelength ranges as observed in Figure 5, but it can be clearly appreciated in the 1089–1325 nm wavelength range, where most of the experiments were executed in the values close to $1 \times 10^6$, where the training times were low, with most of them less than 100 s. For the 900–1700 nm wavelength range, the C values with higher scores may be assumed close to 0–5000 as shown by the darker red color in the left-bottom from the same Figure. For the next 1239–1353 nm wavelength range, there are two possible values of C; the first is between 0 and 5000 and the second is between 650,000 and 800,000. However, the second one should be considered, because of the shorter training times. Finally, for the 1422–1583 nm wavelength range, the C values to be selected are close to 200,000 due to the low average training times and high Accuracy and F2-score.

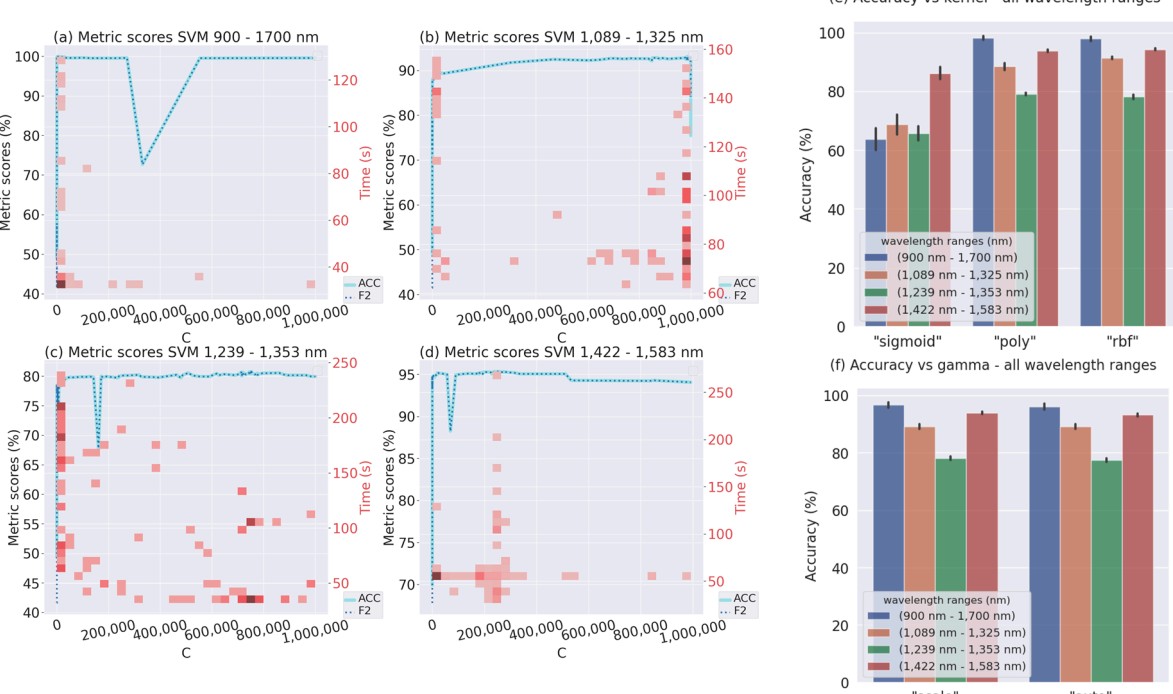

**Figure 5.** (**a–d**) show the Accuracy, F2-score, and time vs. the C hyperparameter. (**e**,**f**) show the accuracy vs. the kernel and gamma hyperparameters.

As seen in Figure 5e,f the sigmoid kernel did not work well, therefore, it is best to choose either the poly or rbf kernels, due to the high Accuracies sometimes exceeding 90%. For the gamma, both types are eligible since the Accuracy result is similar for all wavelength ranges.

### 3.1.2. XGBoost

Four hyperparameters were considered for this study: alpha is the L1 regularization; lambda is the L2 regularization term on weights; max_depth is the maximum depth of a tree; and num_round is the number of rounds for the boosting. The ranges of the parameters used for the tuning are shown in Table 4.

**Table 4.** Search space of the XGBoost method.

| Hyperparameters | Lower Limit | Upper Limit |
| :---: | :---: | :---: |
| alpha | 0 | 1000 |
| lambda | 0 | 1000 |
| max_depth | 0 | 10 |
| num_round | 1 | 4000 |

Figure 6 shows the Accuracy and F2-score for the model with higher performance (from 10 repetitions) generated by the Bayesian optimization. In this case, we only included the graph for one wavelength range (1422–1583 nm), since the others were similar, to analyze the four numerical hyperparameters and avoid making the analysis extensive. The maximum Accuracy and F2-score were 95%.

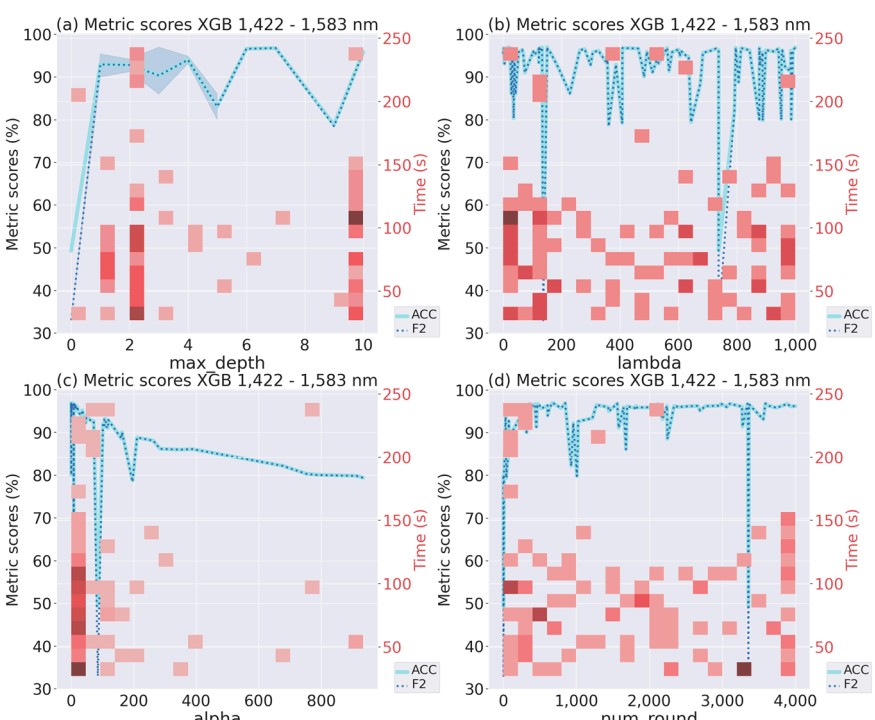

**Figure 6.** Hyperparameters of the best XGBoost model in the 1422–1583 nm wavelength range.

In the case of max_depth, the best results values may be assumed close to 2–4 or 10; as shown by the darker red color and the blue line in Figure 6a. Furthermore, the training time is low and stays under 120 s for most of the values. In the case of lambda, there is no tendency, so many values can be selected. However, as can be seen in Figure 6b, there is a slightly greater proportion of darker tones around the values 0–200, with shorter training times (under 70 s). Hence, these values could be selected. For the case of alpha, it is possible to observe a tendency in the results; hence, as long alpha approaches zero, the F2-score and Accuracy increase. Finally, like for lambda, while many values can be selected for the num_round, the ones around 0–500 or 3000–4000 could be chosen due to the high repetition of experiments.

Table 5 shows the hyperparameter selected by the Bayesian optimization method. As can be seen, the hyperparameters selected coincide with the best values analyzed in Figures 5 and 6 for both ML methods in all the wavelength ranges.

**Table 5.** Hyperparameters selected for the ML methods by applying the Bayesian optimization method.

| Model | 900–1700 nm | 1089–1325 nm | 1239–1353 nm | 1422–1583 nm |
|---|---|---|---|---|
| SVM | 'C':3732.752, 'gamma': "scale", 'kernel': "rbf" | 'C': 985,957.277, 'gamma': "scale", 'kernel': "rbf" | 'C': 752,630.194, 'gamma': "auto", 'kernel': "poly" | 'C': 237,515.537, 'gamma': "auto", 'kernel': "rbf" |
| XGBoost | 'alpha': 0.0, 'lambda': 0.0, 'max_depth': 6, 'num_round': 729 | 'alpha': 0.0128, 'lambda': 0.982, 'max_depth': 10, 'num_round': 1599 | 'alpha': 1.283, 'lambda': 60.669, 'max_depth': 8, 'num_round': 142 | 'alpha': 0.0, 'lambda': 150.466, 'max_depth': 3, 'num_round': 4000 |

### 3.2. Deep Learning Hyperparameter Tuning

For the hyperparameter tuning of the DNNs, we used the proposed methodology explained in Section 2.4.3.4. Figure 7 shows the structure, the hyperparameter values, and the other parameters used for the tuning methodology.

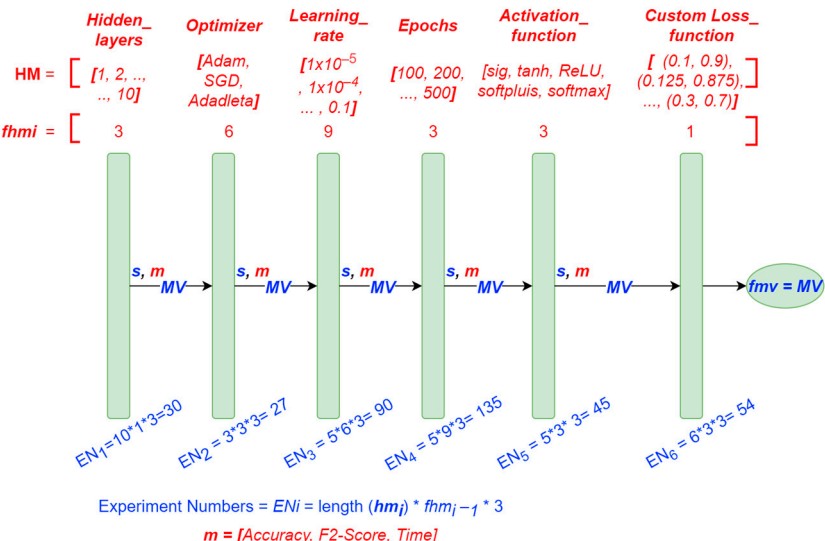

**Figure 7.** Graph procedure of the proposed tuning methodology and the values used.

As shown in Figure 7, the order followed during the tuning methodology was: (1) Hidden layers, (2) Optimizer, (3) Learning rate, (4) Epochs, (5) Activation function, and (6) Custom loss function. The metrics used to rate the experiments during each $hm_i$ iteration were Accuracy, F2-score, and training time. The total number of experiments executed was 381. The tuning methodology implemented in Python allows the user to select the order of the hyperparameters, which we selected considering some studies in the literature and after executing a few experiments. We only included the most relevant experiments; otherwise, the study would have turned out to be very long.

The number of hidden layers was the first hyperparameter we selected to be tuned by the methodology. Secondly, the optimizer and learning rate was selected based on [47], which suggests tuning the learning rate first could save a lot of experiments. Subsequently, the epochs were tuned to adjust the underfitting or overfitting models. The activation function was next selected. At this point, expecting to have a model with good performance, we selected a custom loss function and slightly modified it to maximize the FN, considering the importance of these values in the prediction of the presence of gluten, as explained in Section 2.4.4.

- Hidden layers

We did a few experiments and realized that when increasing the hidden layer, while the performance did not see major changes, the time increased considerably. This suggests that our binary classification problem can be solved without using hundreds of hidden layers. Furthermore, in a literature review was reported that among different studies good results were obtained by using no more than 10 hidden layers [48]. Therefore, we provided the tuning methodology with a vector from 1 to 10 and all the hidden layers with 10 neurons each. Figure 8 shows the score metrics obtained in the $hm_1$ iteration by the hidden layers hyperparameter in the four wavelength ranges.

The first aspect to highlight from Figure 8 is that if the number of hidden layers increases the training time increases, as expected. Regarding the Accuracy and F2-score, in the 900–1700 nm and 1089–1325 nm wavelength ranges, the higher values are located between 6 and 8 hidden layers, but the scores, in general, are very similar. However, In the 1422–1583 nm wavelength range, if the number of hidden layers increases, the Accuracy and F2-score slightly increase.

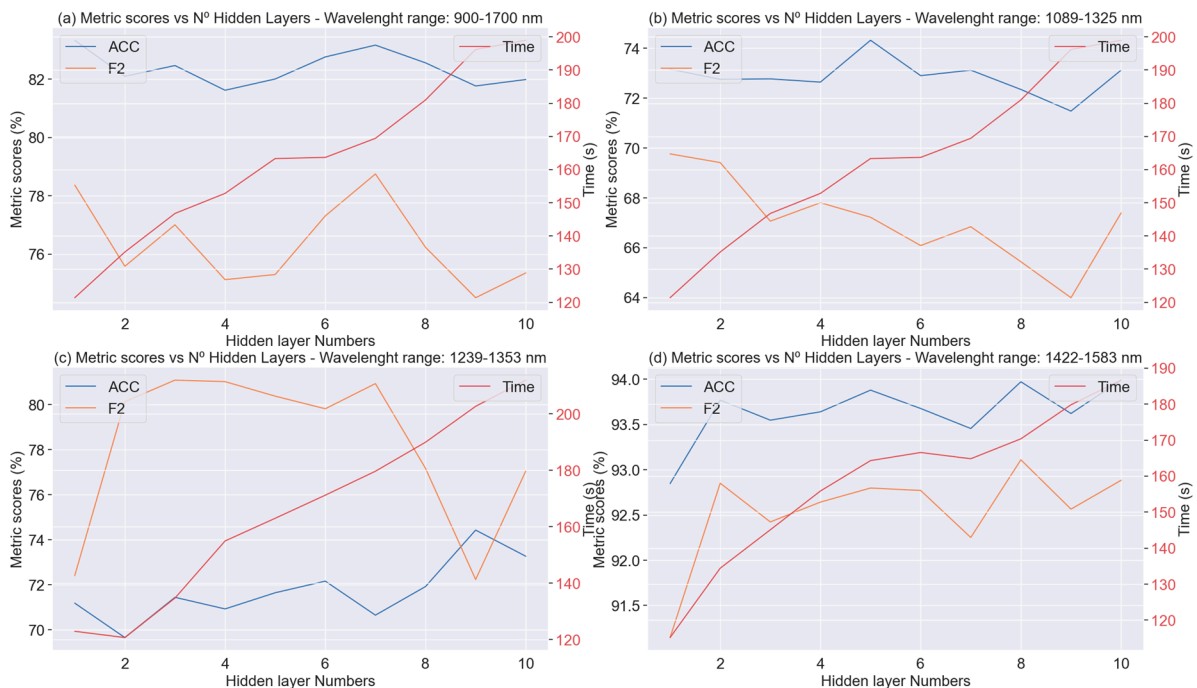

**Figure 8.** Accuracy, F2-Score, Training time vs. Hidden Layers.

- Optimizer

    We selected three different optimizers with the default learning rates. They were tuned taking into account the hidden layers selected in the $hm_1$ iteration.

    Figure 9 shows the experiments developed for all the wavelength ranges using three different optimizers: SGD, Adam, and Adadelta. The black lines located on the top of the bars correspond to the error with a 95% confidence interval. This Figure makes it very clear that the Adam optimizer did not work well and obtained results under 60% for Accuracy and under 50% for F2-score. It can also be noted that the results for SGD and Adadelta are similar. Regarding wavelength ranges, the 2 higher results are for the 900–1700 nm and 1422–1583 nm wavelength ranges, with slightly better performance for the first one.

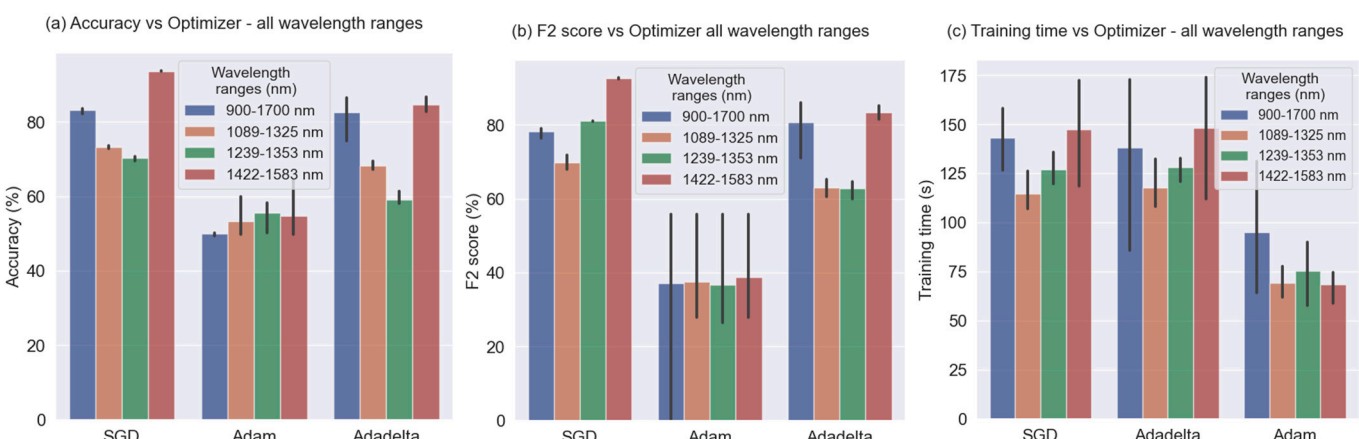

**Figure 9.** Accuracy, F2-Score, and Training time vs. Optimizer.

- Learning rate

    Once the proposed tuning methodology selected the SGD and Adadelta optimizer in the $hm_2$ iteration, 5 values starting from $1 \times 10^{-5}$ to 0.1 were evaluated. Figure 10 shows the Accuracy and training time obtained for each wavelength range. We did not graph

the F2-score because it overlapped with the Accuracy in the shadow zones, making the visualization difficult. The shadow zones represent the mean and 95% confidence interval.

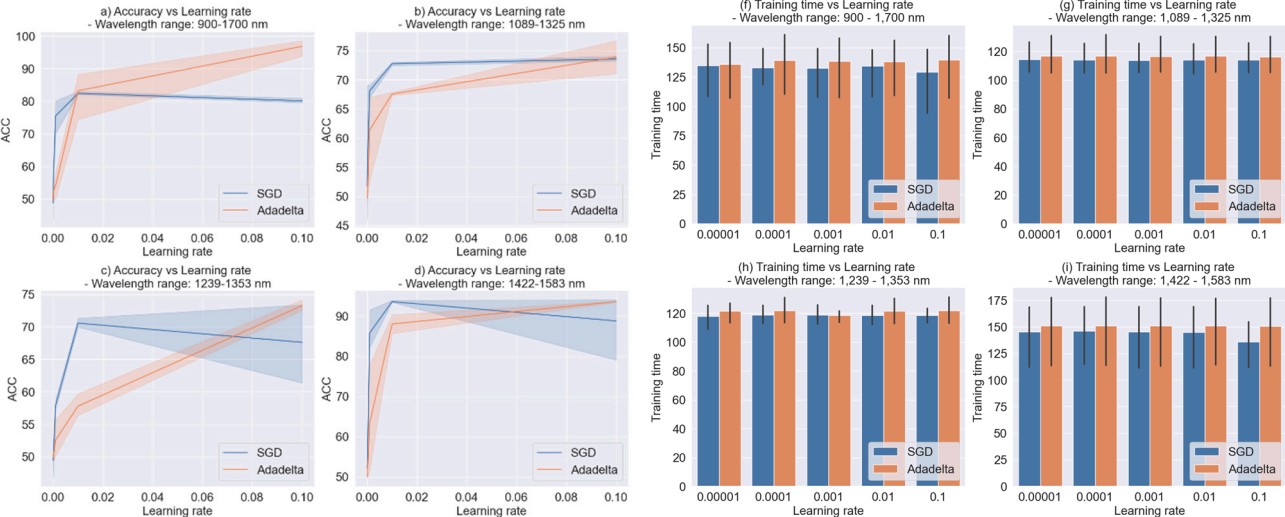

**Figure 10.** Accuracy and Training time vs. Learning rate.

The first aspect to mention is that the Accuracy scores observed in Figure 10 are higher than the ones obtained in the previous iterations $hm_1$ (hidden layers tuned) and $hm_2$ (optimizer tuned), achieving Accuracies and F2-scores over 90% in the 900–1700 nm and 1422–1583 nm wavelength ranges. Thus, the improvements in performance are remarkable for each wavelength range. Another aspect to highlight is that the variability in the SGD optimizer was higher than the Adadelta optimizer, increasing when the learning rate values were higher as seen in Figure 10c,d. The conclusion is that higher results were obtained for the Adadelta optimizer with high values for the learning rate. For the training time, there are no remarkable differences in comparison with the previous $hm_i$ iterations.

- Epochs

All the previous $hm_i$ iterations were executed using 100 epochs. When checking the 135 experiments for the epochs hyperparameter, we observed that most of the models were underfitting and a few other models needed more epochs to finish the learning. It is worth clarifying that all these graphs were obtained at the end of the training by the proposed tuning methodology, therefore, there may be existing underfitting and overfitting models. Unfortunately, this is an aspect that our methodology cannot avoid as we did not consider the loss curves in the metrics $m$ to rate the models.

Figure 11 shows the result of increasing the epochs for the 900–1700 nm wavelength range, the only one we included because the results of the others were very similar. As can be seen in this Figure, when increasing the epochs, the Accuracy, F2-score, and stability of the graphs improve. However, if this value is highly increased, for example in the case of 400 epochs, the model will be overfitting.

- Activation function

The next value to tune was the activation function. The higher results were obtained with the tanh activation function for all the wavelength ranges, therefore, there are no remarkable differences to show, because all previous experiments were executed in the $hm_i$ iterations already used this activation function.

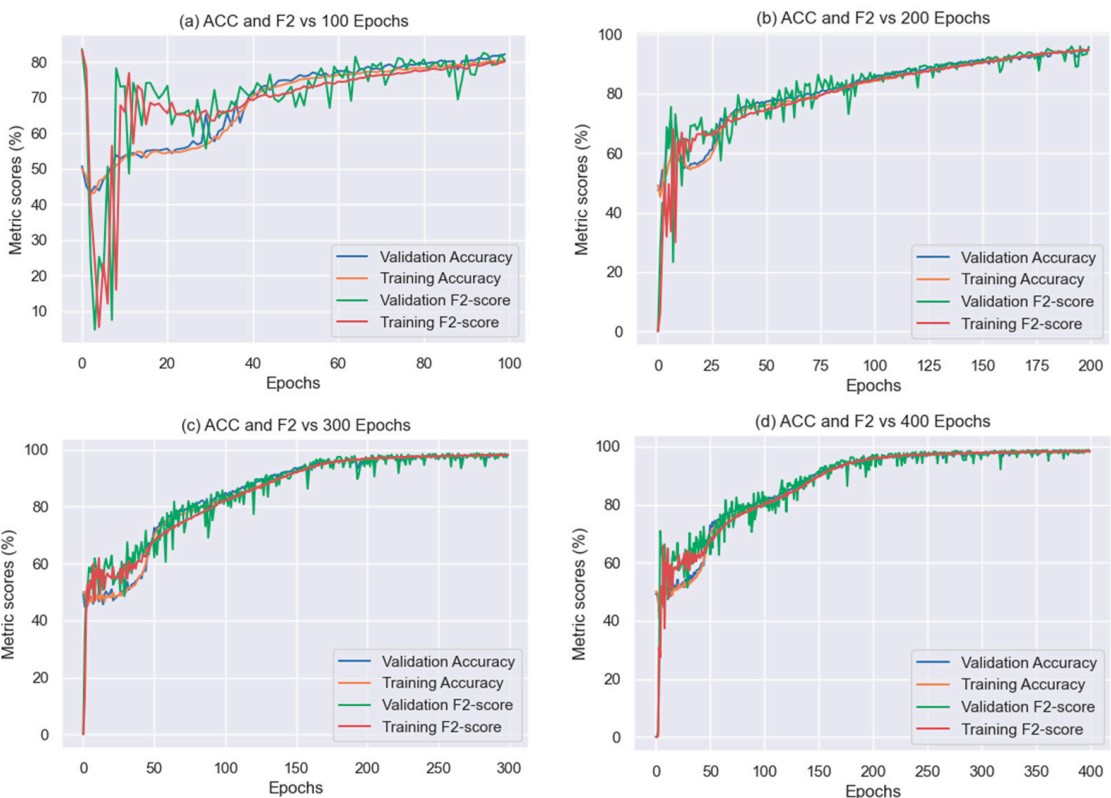

**Figure 11.** Accuracy and F2-score vs. epochs.

- Loss function

At this point, expecting to have a model with good performance, we selected a custom loss function and slightly modified it to maximize the FN. To modify the loss function, we only introduced the weights α and β to the binary cross entropy equation as shown in (16).

$$\text{Loss} = -\frac{1}{n}\sum_{i=1}^{n}\left(\alpha Y_i \cdot \log \hat{Y}_i + \beta(1 - Y_i)\cdot \log(1 - \hat{Y}_i)\right) \tag{16}$$

The above equation gives more weight to the predictions of FN than FP. After a few experiments, we realized that the loss function worked well for values of β over 0.7 and α = 1 − β. Therefore, we executed some experiments around these values which are shown in Figure 12.

On the one hand, if the value of β is very high, then the FN has more importance in the prediction, as seen in the F2-score in Figure 12. On the other hand, when β starts decreasing the F2-score and Accuracy are almost the same. Thus, it is necessary to obtain a tradeoff of how much the FN can be maximized without losing too many prediction rates for the FP.

- Summary of the proposed tuning methodology

We included a summary of the whole procedure in one graph, including the loss function, Accuracy, and F2-score graphs.

Figure 13 shows the higher results for the 6 $hm_i$ iterations for the 900–1700 nm wavelength range. In the $hm_1$ iteration (hidden layers tuned), it can be noted that the F2-score and the loss were good during the training process after tuning the hidden layers, but very unstable for the validation process. Subsequently, in the $hm_2$ iteration (optimizer tuned) the F2-score and loss improved the stability, and while the score of the former slightly increased, that of the latter slightly decreased. In the $hm_3$ iteration (learning rate tuned), there was a remarkable increase in the F2-score. Afterward, in the $hm_4$ iteration (epochs tuned), the F2-score and the loss function improved, with the latter continuously declining.

However, the model seems to be a little overfitting. In the **$hm_5$** iteration (activation function tuned), there is no difference with **$hm_4$**. Finally, in the **$hm_6$** iteration (loss function tuned), where the custom loss function was used, it can be appreciated that the F2-score is slightly higher than the Accuracy.

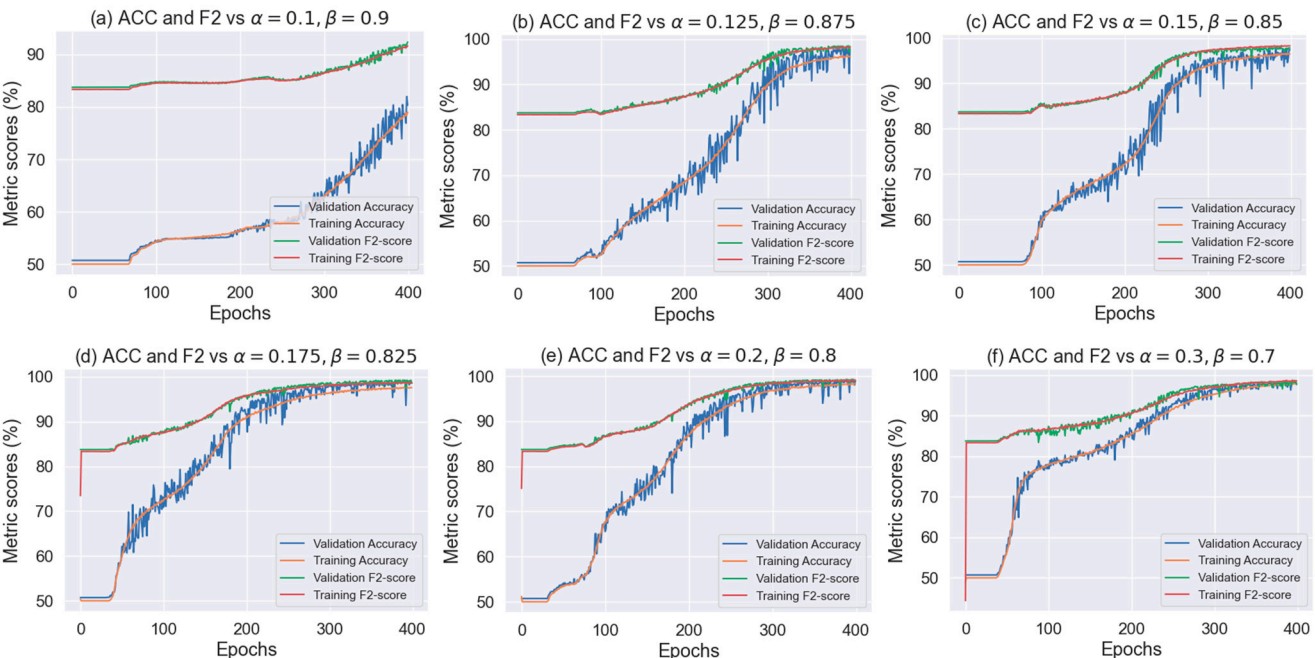

**Figure 12.** Accuracy, F2-score, and loss vs. $\alpha$ and $\beta$ loss function parameters.

Table 6 shows the hyperparameters selected for all the wavelength ranges after completing the proposed tuning methodology. It can be appreciated that the hyperparameters are similar for all the wavelength ranges, having the same optimizer and activation function and almost a similar learning rate. The hidden layers and loss function have different values for each wavelength range.

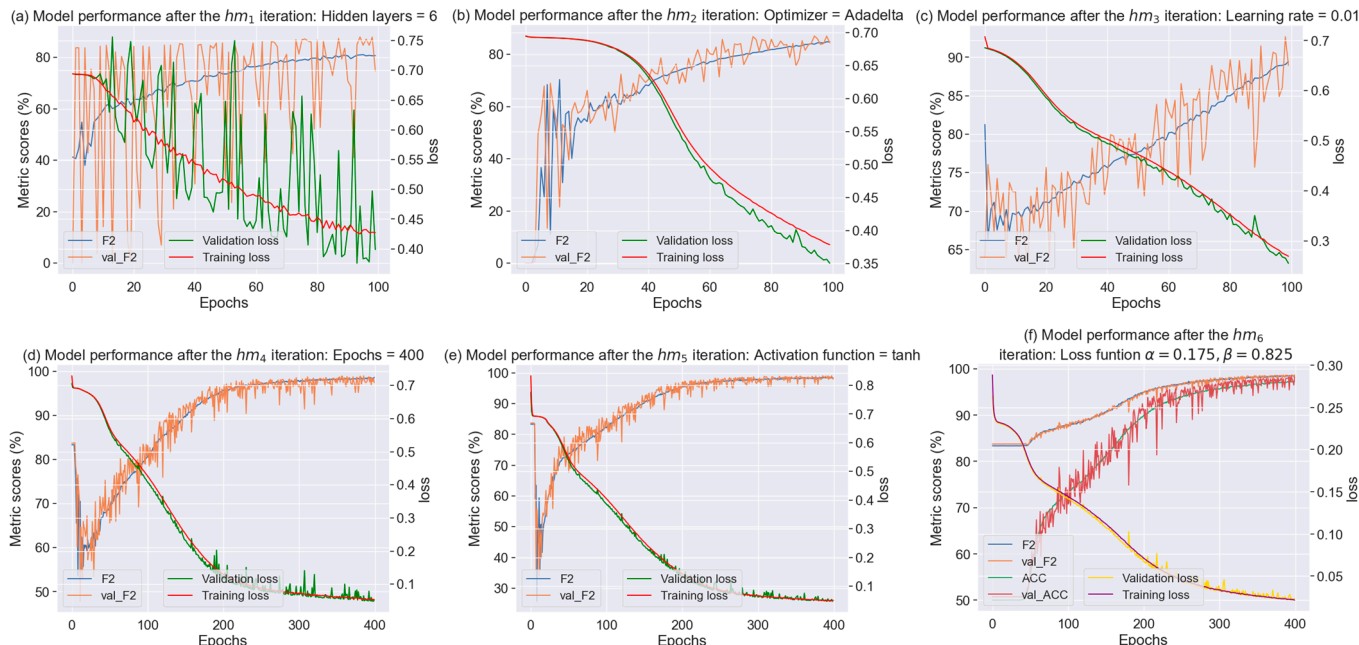

**Figure 13.** Summary of the proposed tuning methodology.

**Table 6.** The hyperparameters selected after the proposed tuning methodology.

| Hyperparameter | Hidden Layers | Optimizer | Learning Rate | Epochs | Activation Function | Loss Function |
|---|---|---|---|---|---|---|
| **900–1700 nm** | 6 | Adadelta | 0.01 | 400 | tanh | $\alpha = 0.175$ $\beta = 0.825$ |
| **1089–1325 nm** | 4 | Adadelta | 0.1 | 500 | tanh | $\alpha = 0.2$ $\beta = 0.8$ |
| **1239–1353 nm** | 3 | Adadelta | 0.1 | 300 | tanh | $\alpha = 0.3$ $\beta = 0.7$ |
| **1422–1583 nm** | 8 | Adadelta | 0.1 | 500 | tanh | $\alpha = 0.15$ $\beta = 0.85$ |

### 3.3. Classification Results

Table 7 shows the Accuracy and F2-score with the higher results for the SVM and XGBoost methods for each wavelength range evaluated. The metric scores were obtained by predicting the testing set which contained 1808 observations of flour samples of rye, corn, and oats, as detailed in Section 2.4.1. The models used the hyperparameters from Table 5.

**Table 7.** Classification results of the ML models.

| Model | 900–1700 nm | 1089–1325 nm | 1239–1353 nm | 1422–1583 nm |
|---|---|---|---|---|
| SVM | ACC = 0.9131 F2 = 0.9445 TT = 72 s | ACC = 0.7863 F2 = 0.8966 TT = 117 s | ACC = 0.7814 F2 = 0.8963 TT = 128 s | ACC = 0.8893 F2 = 0.8550 TT = 97 s |
| XGBoost | ACC = 0.7769 F2 = 0.8675 TT = 383 s | ACC = 0.7625 F2 = 0.8603 TT = 854 s | ACC = 0.5755 F2 = 0.6658 TT = 143 s | ACC = 0.9452 F2 = 0.9287 TT = 1026 s |

ACC, Accuracy; F2, F2-score; TT, Training time.

In Table 7, it can be appreciated that the best model for SVM was in the 1422–1583 nm wavelength range, achieving an Accuracy of 88.93% and an F2-score of 89.28%. It is worth mentioning that, despite the SVM reaching an Accuracy of 91.31% and an F2-score of 90.79% in the range of 900–1700 nm wavelength range, this model is not considered the best one because it used all the features during the training and could be overfitting, as seen in Figure 6a. The higher and the best results among the ML models were obtained by the XGBoost model in the 1422–1583 nm wavelength range, achieving an Accuracy of 94.52% and an F2-score of 92.87%. This model also has the advantage of only using 50 features for the training.

Table 8 shows the classification results of the DNN models with higher scores (1st row) and the DNN models after completing the $hm_6$ iteration of the proposed tuning methodology (2nd row). The testing set evaluated was the same as for the ML methods.

**Table 8.** Classification results for the DNN model.

| Model | 900–1700 nm | 1089–1325 nm | 1239–1353 nm | 1422–1583 nm |
|---|---|---|---|---|
| Higher classification results for DNN | ACC = 0.9503 F2 = 0.9447 TT = 575.8121 s | ACC = 0.7089 F2 = 0.8936 TT = 586.9577 s | ACC = 0.7020 F2 = 0.8998 TT = 582.0159 s | ACC = 0.9177 F2 = 0.9606 TT = 575.8121 s |
| DNN after completing the tuning methodology | ACC = 0.9064 F2 = 0.9370 TT = 521.2503 s | ACC = 0.7089 F2 = 0.8936 TT = 586.9577 s | ACC = 0.7020 F2 = 0.8998 TT = 582.0159 s | ACC = 0.9177 F2 = 0.9606 TT = 766.5958 s |

ACC, Accuracy; F2, F2-score; TT, Training time.

As seen in Table 8, the higher Accuracy and F2-score are in the 900–1700 nm and 1422–1583 nm wavelength ranges. For the 1089–1325 nm, 1239–1353 nm, and 1422–1583 nm wavelength ranges, the higher Accuracy and F2-score results (1st row) are the same as those obtained after completing the tuning methodology (2nd row). However, for the 900–1700 nm wavelength range, the best Accuracy and F2-score (listed in the first row) were achieved in the $hm_4$ iteration (epochs tuned), therefore, the last 2 $hm_i$ iterations did not contribute to improving the model.

The training time for most of the experiments was higher for the DNN in comparison to the ML methods, as expected. However, this does not pose a problem for the prediction of the presence of gluten in new samples because the models are already trained, and the forward propagation time was <1 s for all the experiments. This is very positive for the implementation of a rapid gluten detection technique.

Finally, considering the ML and DL experiments, **the best model was the DNN in the 1422–1583 nm wavelength range**, achieving an **Accuracy** of **91.77%** and an **F2-score of 96.06%** in the prediction of the presence or absence of gluten in three different types of flour.

## 4. Discussion and Conclusions

This study presents a rapid, innovative, budget-friendly, and accurate IoT solution to predict the presence (doped with 100 g/kg + gluten naturally contained) and the absence (0 g/kg + gluten naturally contained) of gluten in 3 different types of flour samples (rye, corn, and oats). The development of the IoT prototype required a difficult and lengthy process comprised of data collection, the development of a serverless architecture for storing and data analysis, the creation of AI models to make predictions, and the visualization of the results. The results section showed very optimistic results in terms of performance metrics in the prediction of the absence or presence of gluten. The best DL model obtained 91.77% for Accuracy and 96.06% for F2-score, and similarly, among the ML models, the best result was obtained by the XGBoost model achieving an Accuracy of 94.52% and F2-score of 92.87%. Both models achieved high performances; however, we selected the DL models because we want to prioritize the F2-score giving more importance to the FN. This was done with the users of the prototype in mind; for whom it is more important to identify when the prototype wrongly predicts the samples as lacking gluten (output of the DNN: zero) when they actually contain it. For instance, in a real-life scenario, the wrong prediction of the absence of gluten in the flour would cause the user to consume gluten and suffer from potential adverse health effects. While the FPs also are important, the erroneous prediction of the presence of gluten does not pose a risk to the user, as he/she would simply not eat flour that is in reality gluten-free.

Regarding the proposed tuning methodology for the DNNs, one aspect to highlight is that we achieved good results executing a low number of experiments (327), given the total of possible combinations (3750) of the 28 hyperparameters evaluated. Furthermore, the total number of experiments could be reduced to 109 without repetition in the DNNs training.

We cannot make a fair comparison in terms of Accuracy and F2-score between our results and those of other studies due to the difference in the evaluation metrics and analytical methods employed. While other studies make use of instrumental analyses such as chromatography or enzyme immunoassay as ELISA, direct methodologies where the analyte (protein) is the target to be measured, in our approach (NIRS + AI) we used the data obtained from electromagnetic spectrum and emitted by food matrix to quantify the target analyte indirectly. Nevertheless, we can highlight the differences in the execution in terms of time and complexity of the experiment. Regarding the time needed to make a prediction, our IoT prototype portable solution provided optimistic results with the DL and ML models needing less than 1 s to predict an observation. Together with the duration of the collection in the platform (measuring time + storing time), which varies between 30–60 s, the time needed to predict the presence of gluten in any of the new rye, corn, and oat flour samples of ≈400 mg adds up to <1 min. Regarding the training time,

the best DNN model needed ≈575.8121 s to complete the training. However, this will not be an issue for the final application because the models would be trained already. In terms of the complexity of the experiment, our IoT prototype portable solution also has a great ease-of-use advantage, because the user only needs the kit (sensor, Raspberry Pi 4, DLPNIRNANOEVM sensor, 3D Mechanical system, and collecting plate) and a device with an internet connection (mobile, tablet or pc), and to put the samples into the collecting plate and press a few clicks to make the prediction. Instead, other analytical methods take significantly more time, counting from the sample preparation until the data recovery; around 0.5–2.5 h for ELISA [13]; ≥0.5 h for HPLC [49]; and > 3 h for PCR [50]. Other studies have achieved very good results while decreasing the detection time, to illustrate, a recent study designed an optical nanosensor for rapid detection of the gluten content of samples containing wheat. They obtained results for the determination coefficient (R2) with 0.995 for folic acid-based-carbon dots molecularly imprinted polymer (FA-CDs-MIP) and 0.903 for FA-CDs none-imprinting polymer (FA-CDs-NIP), within a range of gluten detection of 0.36 to 2.20 μM [51]. They reported less than 4 min as the response time for the florescent nanosensor. In another investigation, a real-time artificial intelligence-based method was employed to detect adulterated lentil flour samples that contained trace levels of wheat (gluten) or pistachios (nuts) [19]. The authors used images taken with a simple reflex NIKON camera, model D5100, to train the network, with a total of 2200 images collected in a well-lit room without any spotlight illuminating the samples. Despite the difference in the input data and food samples employed, this study is one of the most similar to ours, as it achieved an accuracy of 96.4% in the classification of wheat flour, showing the potential of the NIRS technology. At first, the method seems to be accurate and particularly quick, with the photo capturing and the prediction of the CNN networks being instantaneous (the authors mention seconds). However, in contrast with our study, it raises several concerns regarding its implementation in an environment production, such as the difficult setup when using a big camera, the users involved, and the changing environmental conditions depending on the light to which the samples are exposed. Other authors conducted a multiclass classification study using ML approaches and the Fourier-Tranform (FTIR) Spectroscopy to detect and quantify cross-contact gluten in flour [52]. The samples used were non-gluten (corn flour) and gluten flours (wheat, barley, and rye) with a total of 640 samples (200 × 3 for contaminated samples and 10 × 4 pure samples). The supervised classification method was the Partial Least Square Discriminant Analysis (PLSDA) achieving true positive rates (TPR) of 0.87500, 0.81250, 0.9333, and 1.0 respectively for barley flour, wheat, rye, and corn flour classes. While the analytical method employed was different, the study showed optimistic results in addressing problems of multiclassification and quantification of gluten in the flour, a topic we plan to investigate in our future research. With regards to the time and resources employed, the authors stated that the study was developed in non-real time and that the samples were prepared using the FTIR spectrometer (Nicolet iS 50 Waltham, MA, USA, North America), laboratory equipment that due to its nature presents limitations, such as the need for user training and the difficult transportation among others. This highlights the value of the method employed in our study, which can be easily taken anywhere by the user and does not require any specific training. Finally, with the best results achieved in the present study (the DNN model in the 1422–1583 nm wavelength range), we improved the results achieved in our previous study, where we employed the Random Forest classifier [31], overcoming the Accuracy by 5% and the F2-score by 9%. It is worth mentioning that in that study the main objective was to find the best wavelength ranges using feature selection techniques.

Our study suffers from several limitations. First, it is only able to classify the presence or absence of gluten considering samples of 0 g/kg and 100 g/kg of gluten concentration. Therefore, the IoT prototype portable solution has a high chance of failure with middle concentrations samples, for example, 50 g/kg, while other experiments can quantify the gluten in samples under 100 mg/kg by using highly sensitive methods, such as enzyme immunoassay techniques or instrumental analytical techniques (capillary electrophoresis,

PCR, QC-PCR, RP-HPLC, LC-MS, and MALDI-TOF-MS). However, these methods also imply elevated costs and specialized training [12,13,15]. While our prototype could not be considered low budget, with the overall costs amounting to 1200 euros, it is still less expensive than other methods [53–57] (around 1000–17,000 euros) and does not require specialized training. The methodology employed in our study has other limitations. First, few previous experiments are required to know the order of the hyperparameters. For example, in this study, we used the tanh activation function from the $hm_i$ iteration, because it showed to have better performance of Accuracy than other activation functions. It is worth mentioning that these experiments are not executed to obtain higher or the best results, but they are useful to choose the initial hyperparameters. The lack of an optimal code implementation in our proposed tuning methodology constitutes another limitation, making it not possible to perform a fair comparison with other existing methods, such as Random Search, Bayesian Hyperparameter tuning, or Grid Search, since it ignores aspects such as vectorization and modular programming.

When it comes to future steps, we are considering working first on a multiclass problem that allows classifying the content of gluten in different levels and applying further DL and ML regression, which lets us quantify the amount of gluten in the samples examined. We also want to implement the proposed tuning methodology as a full Python framework, making it possible to validate it with different datasets and to compare the performance with other existing tuning hyperparameters frameworks.

The binary classification results obtained in this study are promising for the future designing of real-time IoT devices that allow rapid detection of the presence or absence of gluten and can be used by any ordinary person. Therefore, this study contributes to the state of the art of NIRS + Artificial Intelligence applied in the food industry and is aligned with the requirements of the 4.0 industry.

**Author Contributions:** Conceptualization, O.J.-B.; methodology, O.J.-B. and A.O.S.; software, O.J.-B. and A.O.S.; validation, O.J.-B., A.O.S., L.B.-L. and B.G.-Z.; formal analysis, O.J.-B., A.O.S., L.B.-L. and B.G.-Z.; investigation, O.J.-B. and A.O.S.; resources, L.B.-L.; data curation, O.J.-B. and A.O.S.; writing—original draft preparation, O.J.-B. and A.O.S.; writing—review and editing, O.J.-B., A.O.S., L.B.-L. and B.G.-Z.; visualization, O.J.-B. and A.O.S.; supervision, B.G.-Z.; project administration, L.B.-L. and B.G.-Z.; funding acquisition, L.B.-L. and B.G.-Z. All authors have read and agreed to the published version of the manuscript.

**Funding:** This research was funded by the Department of Economic Development, Sustainability and Environment of the Basque Government, grant number KK-2021/00035 (ELKARTEK), and eVIDA research group IT1536-22.

**Data Availability Statement:** Data are contained within the article.

**Acknowledgments:** The authors would like to thank A. Olaneta and L. Zudaire from Leartiker S. Coop for their support in the preparation and analysis of the flour samples and the literature review. Likewise, to Unai Sainz Lugarezaresti, Guillermo Yedra Doria, and Uxue García Ugarte from the eVIDA Lab of the University of Deusto for the support in the development of the 3D mechanical system and the data collection.

**Conflicts of Interest:** The authors have no conflict of interest to declare.

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
