# Peer review of "IoT System for Gluten Prediction in Flour Samples Using NIRS Technology, Deep and Machine Learning Techniques"

_electronics, doi:10.3390/electronics12081916_

Round 1

Reviewer 1 Report

The authors used IoT technology based on the machine learning and deep learning   approaches for prediction   gluten. The article very interested for food security.

1-   literature review very poor authors need to add some recent references

2-    Why authors not used evaluation metrics like specificity and recall

3-   Where is figures of confusion metrics

4-   The quality of figures very poor authors need to improve the quality of figures

5-   How many classes on dataset?

6-    I am not finding numbers of sample of dataset

7-   Comparison between the proposed system and existing is not variable, author should compare system result with recent existing systems

8-   In conclusion, need to add limitation and future scope of this study   

Reviewer 2 Report

I believe that the article was very carefully written. The results presented in the manuscript are helpful and promising. However, I have a few comments and asks the authors.

1.       Line 77,80,83. change the way you cite in whole manuscript

2.       Would the samples of rice, corn and oats wheat used in the tests be marked as "gluten-free"?? Did the authors treat these samples as naturally gluten-free?

3.       The descriptions in the drawings are not very clear

4.       The gluten level results in the last column of Table 1 are unclear to me. Is it a natural level of gluten or after adding it? This is not clear in the text, as the authors refer twice to Tebel 1 in different contexts. In the description of the table, it should be added whether the gluten level applies to samples before or after fortification.

5.       please indicate the type of ELISA test used for the study

Reviewer 3 Report

The article describes an IoT system for gluten prediction in flour samples using data acquired with infrared sensors and processed with deep and machine learning algorithms.

Although the idea of article is not a new one, its content represents an improvement in quantitative and qualitative aspects in comparison with previous work made mostly by the same authors.

Unfortunately the entire article in an inappropriate manner, many results (figures) are spreaded all over the entire results section, making the article difficult to understand and follow.    

A comparative analysis of the results obtained vs similar approaches is missing from the discussion and conclusions section.   

More details about the HW part of the system are necessary, mainly for the internet interfaces and data transfer.

Round 2

Reviewer 1 Report

Authors addressed my comments

Reviewer 3 Report

The raised issues have been addressed by the authors and solved properly. In these circumstances I recommend the publication of the article in its actual form.